# Dietary Microplastic Administration during Zebrafish (*Danio rerio*) Development: A Comprehensive and Comparative Study between Larval and Juvenile Stages

**DOI:** 10.3390/ani13142256

**Published:** 2023-07-10

**Authors:** Nico Cattaneo, Matteo Zarantoniello, Federico Conti, Andrea Frontini, Giulia Chemello, Beniamino Dimichino, Fabio Marongiu, Gloriana Cardinaletti, Giorgia Gioacchini, Ike Olivotto

**Affiliations:** 1Department of Life and Environmental Sciences, Marche Polytechnic University, 60131 Ancona, Italy; n.cattaneo@pm.univpm.it (N.C.); f.conti@pm.univpm.it (F.C.); a.frontini@univpm.it (A.F.); g.chemello@univpm.it (G.C.); beniamino.dimichino@gmail.com (B.D.); fabiomarongiu96@gmail.com (F.M.);; 2Department of Agricultural, Food, Environmental and Animal Sciences, University of Udine, 33100 Udine, Italy; gloriana.cardinaletti@uniud.it

**Keywords:** microplastics, histology, oxidative stress, immune response, fish development

## Abstract

**Simple Summary:**

Microplastics (MPs) contamination is a worldwide problem. Studies have also demonstrated their presence in fish feed, posing serious issues for the aquaculture sector. The present study investigated, for the first time through a comparative approach, the effects of different-sized fluorescent MPs included in a diet intended for zebrafish (*Danio rerio*). A comparison based on fish developmental stage (larval vs. juvenile), exposure time, and dietary MPs’ size and concentration was performed, applying a set of laboratory analyses to elucidate MPs’ possible effects on fish growth and welfare, translocation among tissues and organs, and the presence of biological barriers capable of trapping MPs. Results showed that smaller MPs and longer dietary exposure are responsible for translocation of MPs from the gut to other tissues and organs. However, the biological barriers of zebrafish are able to limit MPs’ translocation to the muscle. Results obtained in this experimental model are important for possible application to other farmed finfish species.

**Abstract:**

One of the main sources of MPs contamination in fish farms is aquafeed. The present study investigated, for the first time through a comparative approach, the effects of different-sized fluorescent MPs included in a diet intended for zebrafish (*Danio rerio*). A comparison based on fish developmental stage (larval vs. juvenile), exposure time, and dietary MPs’ size and concentration was performed. Four experimental diets were formulated, starting from the control, by adding fluorescent polymer A (size range 1–5 µm) and B (size range 40–47 µm) at two different concentrations (50 and 500 mg/kg). Zebrafish were sampled at 20 (larval phase) and 60 dpf (juvenile stage). Whole larvae, intestine, liver and muscles of juveniles were collected for the analyses. Polymer A was absorbed at the intestinal level in both larvae and juveniles, while it was evidenced at the hepatic and muscular levels only in juveniles. Hepatic accumulation caused an increase in oxidative stress markers in juveniles, but at the same time significantly reduced the number of MPs able to reach the muscle, representing an efficient barrier against the spread of MPs. Polymer B simply transited through the gut, causing an abrasive effect and an increase in goblet cell abundance in both stages.

## 1. Introduction

Microplastics (MPs; size < 5 mm) have been detected in several environments [1,2,3,4], the marine ecosystem included [4], making them a worldwide threat. MPs are divided into two categories: (i) primary MPs are directly produced for commercial use, such as for cosmetics [5] or industrial abrasives [6]; (ii) secondary MPs are the result of the fragmentation of larger plastic debris through natural weathering processes [7]. In oceans, MPs were discovered decades ago, and it has now been demonstrated that they are present everywhere from the water surface to the sediment [8,9]. MPs have also been detected in marine animals, from the lowest trophic level to the top of the food chain, posing serious concerns about their effects on living organisms [10,11,12].

Different studies have demonstrated that the aquaculture sector is also affected by MPs pollution since farmed aquatic species show similar MPs accumulation with respect to wild specimens [13,14]. In fish farms, environmental contamination through the use of plastic materials is considered a major source of exposure for farmed species [15,16,17]. However, another important source of MPs pollution for farmed fish is represented by the feeds used in aquaculture. In fact, it has been shown that conventional aquafeed ingredients, such as marine [18,19] and plant-derived ones [20], as well as more recent alternatives such as insect meal [21,22,23], are characterized by different degrees of MPs contamination. Specifically, the number of fragments, polymer type, and size of MPs associated with each aquafeed ingredient are related to both the production/collection area [24] and the processing/packaging procedures [19].

In this context, MPs feed contamination is related to the concentration, size, and shape of the polymers, and all these features can have negative effects on the different life cycle stages of farmed fish [25]. The larval stage is one of the most delicate phases of the fish life cycle because of the shift from endogenous to exogenous feeding and the rapid development characterized by deep morphological, anatomical, and behavioral changes, often associated with high mortality [26,27,28]. In addition to this critical physiological step, it has been demonstrated that fish larvae can mistake MPs for zooplankton, resulting in (i) gastrointestinal tract obstruction; (ii) reduction in predatory activity caused by an apparent feeling of satiety; (iii) reduced growth and swimming capacity; (iv) induction of inflammatory responses, mainly at the gut level; and (v) alterations in the structure of the microbiome at the phylum level [29,30]. Exposure to MP-contaminated diets can also have adverse effects on further fish life cycle stages (juveniles and adults) due to both physical and chemical mechanisms, as evidenced in several species, such as gilthead seabream (*Sparus aurata*) [31,32,33], European seabass (*Dicentrachus labrax*) [34], Indian medaka (*Oryzias melastigma*) [35,36], and zebrafish (*Danio rerio*) [37,38,39]. In particular, with regard to the juvenile stage, gilthead seabream exposed to MPs-contaminated diets showed an increase in oxidative stress marker activity [31], increased cellular stress with a higher oxidative stress response in the brain [32], and enhanced activity of antioxidant and pro-inflammatory enzymes [33]. Furthermore, a recent study conducted on European seabass juveniles demonstrated the translocation of dietary MPs (with a size range of 1–5 μm) into muscular tissue, highlighting potential effects on human health [34].

With regard to adult fish, most of the studies have been performed on experimental model fish, which show a shorter life cycle when compared to commercial species. Adult zebrafish exposed to different-sized MPs showed oxidative stress activation in the intestine [39] and gonad impairment [38], as well as a different location and degree of accumulation depending on the MPs’ size [37]. Adult medaka exposed to MPs-contaminated diets showed histological and oxidative stress alterations in gills, livers, testes, and ovaries, leading to reproductive and endocrine disorders [35], as well as an alteration in gut microbiota, hepatic toxicity, and lipid metabolism disorders depending on MPs’ size [36].

By considering the most recent literature on fish exposure to MPs, it is evident that MPs’ toxicity mainly depends on their size, concentration, shape, and chemical features, but also on fish life cycle stage [40,41]. Consequently, a comparative study between different life cycle stages subjected to MPs-contaminated diets is necessary and of great interest. In this regard, zebrafish represent an excellent experimental model, as they are a widely used species in fish nutrition [42] and toxicology studies (including MPs exposure) [43,44,45] and are characterized by a short life cycle compared to other finfish species [42,46]. Nevertheless, due to developmental, anatomical, and behavioral differences among zebrafish life cycle stages, a comparative study of dietary MPs exposure is of interest [47,48,49]. In particular, one of the main features of teleost that determines the transition from the larval stage, characterized by deep and rapid morphological changes, to a fully developed juvenile is the development of a functional and complete digestive system [48].

In the present study, diets containing fluorescent MP microbeads of different size ranges (1–5 µm or 40–47 µm) and included at two different concentrations (50 mg/kg or 500 mg/kg) were formulated, with the first concentration selected according to Zitouni et al. [50] and the second one 10x higher in order to intensify possible effects due to MPs ingestion. MPs-free or -contaminated diets were prepared and provided to zebrafish, from the larval to the juvenile stage, in order to perform a comparative study of their potential accumulation, translocation, and impact on fish welfare and growth. In this regard, a multidisciplinary approach, including, biometry, chemical digestion, histology, confocal microscopy, and molecular biology, was applied in order to have a comprehensive overview of the effects of dietary MPs administration in both larval and juvenile zebrafish.

## 2. Materials and Methods

### 2.1. Ethics

All the experimental procedures involving animals conducted for the present study were performed in accordance with Italian legislation on experimental animals and were approved by the Ethics Committee of the Marche Polytechnic University (Ancona, Italy) (n.3 24/11/2022) and the Italian Ministry of Health (Aut. n. 391/2023-PR). The suffering of the animals was minimized by using an anaesthetic (MS222; Merck KGaA, Darmstadt, Germany).

### 2.2. MPs Features

Two different fluorescent microbeads (MPs) were purchased from Cospheric LLC (Goleta, CA, USA): (i) polymer A: amino formaldehyde polymer (FMV-1.3), with a size range of 1–5 µm and an emission peak of 636 nm when excited at 584 nm; (ii) polymer B: polyethylene (UVPMS-BR-0.995), with a size range of 40–47 µm and an emission peak of 607 nm when excited at 575 nm. Before being included in the preparation of the experimental diets at the above-mentioned concentrations, fluorescent MP microbeads, hydrophobic in their pristine state, were resuspended, according to the company’s technical support suggestion, in a 0.1% tween-80 solution as a surfactant (Merck KGaA) and then rinsed with deionized water three times; low tween-80 concentrations are non-toxic to zebrafish [51].

### 2.3. Production of Experimental Diets

Five test diets were prepared in the pilot feed mill facility available at the Department of Agriculture, Food, Environmental and Animal Science of University of Udine (Italy) starting from the same batches of single ingredients. A control fluorescent MPs-free diet (named Control) was formulated to resemble the proximate composition of a commercial standard diet available for zebrafish (Zebrafeed; Sparos LDA, Olhão, Portugal), in accordance with a previous study performed on zebrafish [52]. The fluorescent MP beads were not added during the preparation process of Control diet, that was checked through confocal microscopy to confirm the absence of contamination; no fluorescence was detected. Four experimental diets containing MPs were prepared by adding, at two different concentrations (mg/kg of feed), fluorescent polymers A or B to the control mixture diet, which was made of the same ingredients from the same batches: (i) 50 and 500 mg/kg of polymer A (diet A50 and A500, respectively); (ii) 50 and 500 mg/kg feed of polymer B (diet B50 and B500, respectively). It has to be pointed out that the authors of the present study aimed to evaluate only the effects of fluorescent MPs voluntary added (polymers A and B), being aware that each dietary ingredient could possess an intrinsic contamination of other non-fluorescent MPs. However, since the basic diet mixture was made by the same ingredients coming from the same batches, the amount of non-fluorescent MPs in all diets is assumed to be the same, and thus the effects observed in fish are related only to the presence of the fluorescent MPs.

In particular, all powdered ingredients used for the production of the test diets were well mixed (GastroNorm 30C1PN, ItaliaGroup Corporate Srl) for 20 min, and then oil and water were added to the mixture to attain the appropriate consistency for pelleting. Water was used to include A or B polymers in the mixture. Pellets were obtained by using a 3 mm die meat grinder, and were dried at 37 °C for 48 h in a ventilated heater and then ground and sieved through a battery of sieves to obtain particles of the right size for fish rearing (for details, see Section 2.5). Diets were subsequently stored in vacuum bags and shipped to the Department of Life and Environmental Sciences, Marche Polytechnic University (Italy).

Feed samples were analysed for dry matter, crude protein, ether extract, and ash contents according to the AOAC [53]. The ingredients, MPs concentrations, and proximate composition of experimental diets are reported in Table 1.

### 2.4. Fish

Zebrafish embryos (wild-type strain AB) were obtained from the broodstock colony of Università Politecnica delle Marche and maintained in a Tecniplast system (Varese, Italy) for 48 h, with a photoperiod of 12 h light and 12 h dark, under optimal water conditions: 28 ± 0.5 °C; pH 7 ± 0.1; ammonia and nitrite concentrations < 0.01 mg/L; and nitrate concentration <10 mg/L. Then, the embryos were collected, selected under a stereomicroscope by collecting those that were correctly developing and not damaged, and randomly divided into five experimental groups (in triplicate) according to the five dietary treatments. A total of 7500 embryos were used.

### 2.5. Experimental Design

After hatching, zebrafish larvae were initially reared in fifteen 20 L tanks (3 tanks per experimental group; 500 larvae per tank) with the same water conditions as the broodstock tanks. Water was gently replaced 10 times a day via a dripping system and the sides of each tank were provided with black panels to reduce light [55]. After 20 days post-fertilization (dpf), the fish of each tank were transferred in 100 L tanks (3 tanks per experimental group) equipped with mechanical and biological filtration (Panaque, Roma, Italy).

Starting from 5 dpf to the end of the experiment (60 dpf), zebrafish were fed experimental diets as follows: (i) Control group, fed a control diet; (ii) A50 group, fed a diet containing 50 mg/kg of polymer A (range size: 1–5 µm); (iii) A500 group, fed a diet containing 500 mg/kg of polymer A (range size: 1–5 µm); (iv) B50 group, fed a diet containing 50 mg/kg of polymer B (range size: 40–47 µm); (v) B500 group, fed a diet containing 500 mg/kg of polymer B (range size: 40–47 µm). The feed particle size was adapted in relation to fish growth during the feeding trial: <100 µm from 5 to 15 dpf, 101–200 µm from 16 to 30 dpf, and 201–400 µm from 31 dpf to the end of the experiment (60 dpf), following the procedures of [54]. Each experimental group was fed at a feeding rate of 3% body weight, divided into two equal amounts (one in the morning and one in the afternoon). In addition, from 5 to 10 dpf, zebrafish larvae in all the experimental groups were fed the rotifers Brachionus plicatilis (5 individual/mL) twice a day according to Zarantoniello et al. [54]. Uneaten feed and dead specimens, if present, were siphoned 30 min after feeding from all the experimental tanks and recorded. The required numbers of fish per tank were sampled, after a lethal dose of MS222 (0.3 g/L): (i) at 20 dpf (larval phase); (ii) at 60 dpf (juvenile stage). Biological samples were collected and properly stored for further analysis. In particular, samples were immediately placed in biopsy cassettes and fixed in 4% paraformaldehyde (PFA) or in Bouin’s solution at 4 °C for confocal microscopy or histological analyses, respectively. Samples for chemical and molecular analyses were placed in 1.5 mL Eppendorf tubes and stored at −20 or −80 °C, respectively. This storage is considered appropriate since the present study is checking only for fluorescent MPs.

### 2.6. Biometry

Ten newly hatched larvae (3 dpf) from each tank (30 per dietary group) were randomly collected to measure the initial body weight (IBW), determined in pools of five larvae each. For the final body weight (FBW) determination, 20 zebrafish larvae and 20 zebrafish juveniles were randomly collected from each tank (60 larvae and 60 juveniles per experimental group) at 20 and 60 dpf, respectively. The specific growth rate (SGR) was calculated for both larvae and juveniles as follows:SGR (% day^−1^) = [(ln FBW − ln IBW)/t] × 100(1)
in which t represents the number of days (17 and 57 for larvae and juveniles, respectively). Finally, the survival rate was calculated at both 20 and 60 dpf for larvae and juveniles, respectively, by removing the dead specimens from the initial number of fish.

### 2.7. Confocal Microscopy

For confocal microscopy analyses, three feed subsamples of each experimental diet were analysed to evaluate the effective presence of MP microbeads. Additionally, 5 whole larvae per tank (15 per experimental group) were collected at 20 dpf, whereas liver, intestine, and muscle samples from 5 juveniles per tank (15 per experimental group) were collected at 60 dpf. Samples were fixed in 4% PFA for 24 h at 4 °C and then stocked at the same temperature in a 1× phosphate-buffered saline (PBS) solution until further processing. Samples were then placed in concave glass slides with a glycerol-PBS solution (90:10) and mounted with a cover slip. The presence of fluorescent MP microbeads in the collected samples was assessed with a Nikon A1R confocal microscope (Nikon Corporation, Tokyo, Japan). Samples were excited with 561/647 nm wavelengths simultaneously and emissions were collected at 615 and 670 nm to visualize, respectively, the MPs (red) or tissue texture (blue). Images were processed with NIS-Element software (version 5.21.00; Nikon).

### 2.8. Chemical Digestion of Samples and MPs Quantification

Three pools of 10 larvae per tank (9 pools per experimental group) collected at 20 dpf, and liver, intestine, and muscle samples from three juveniles per tank (9 per experimental group) collected at 60 dpf, were weighed and digested through a 10% KOH solution according to Chemello et al. [56]. The solution was added to each sample (1:10 *w*/*v* ratio) in glass tubes; samples were then incubated for 48 h at 40 °C. After 48 h, the digestate was filtered through 0.7 µm pore-size fibreglass filters (Whatman GF/A, Merck KGaA) using a vacuum pump connected to a filter funnel. The filters were dried at room temperature and stocked in glass Petri dishes until the analysis. MPs quantification on filters was performed through a Zeiss Axio Imager.A2 (Zeiss, Oberkochen, Germany) using Texas Red (561 nm) and FITC (491 nm) channels. The MPs were manually counted using the ZEN Blue 2.3 software (Zeiss) and acquisition of images was made by the Axiocam 503 digital camera (Zeiss).

### 2.9. Histology

Five whole larvae per tank (15 per experimental group) were collected at 20 dpf, whereas liver and whole intestine samples from 5 juveniles per tank (15 per experimental group) were collected at 60 dpf. All the samples were fixed in Bouin’s solution (Merck KGaA) for 24 h at 4 °C and stocked in a 70% ethanol solution at 4 °C (after being washed three times with 70% ethanol). Then, samples were dehydrated through ethanol solutions (80, 95, and 100%), washed with xylene (Bio-Optica, Milano, Italy), and, finally, embedded in paraffin (Bio-Optica). The solidified paraffin blocks were cut to obtain 5 µm sections using a Leica RM RTS microtome (Leica, Nussloch, Germany). Sections were stained with (i) Mayer haematoxylin and eosin Y (Merck KGaA; H&E) to assess potential alterations in the tissues’ architecture and the eventual occurrence of inflammatory phenomena in both the intestinal tract and the hepatic parenchyma; (ii) Alcian Blue (Bio-Optica) to measure the relative abundance of Alcian blue positive (Ab+) goblet cells (only for sections of whole larvae and juvenile intestines). The evaluation of histological indexes in the intestine was performed on three transversal sections per fish (15 fish per dietary group) collected at a distance from each other of 50 μm. Specifically, for the morphometric evaluation of the height of undamaged and non-oblique mucosal folds, the ZEN 2.3 software (Zeiss) was used. Regarding the semi-quantitative analysis of the relative abundance of Ab+ goblet cell, scores were assigned as follows: Ab+ goblet cells: + = 0 to 3 per villus; ++ = 4 to 6 per villus; + + + = more than 6 per villus. 

### 2.10. Molecular Analyses

Three larvae per tank (9 larvae per experimental group) were collected at 20 dpf, and liver and whole intestine samples from 3 juveniles per tank (9 liver and 9 intestine samples per experimental group) were collected at 60 dpf. Total RNA extraction was performed using the TRI Reagent (Merck KGaA) and eluted in 20 µL of RNase-free water (Qiagen, Hilden, Germany). DNase treatment (10 IU at 37 °C for 10 min, MBI Fermentas, Milano, Italy) was performed on total RNA to digest genomic DNA. The final RNA concentration/integrity was measured using a NanoPhotometer P-Class (Implen, München, Germany) and by running 1 µg of total RNA stained with GelRed^TM^ on a 1% agarose gel. RNA samples were stored at −80 °C. The cDNA synthesis was performed on 1 µg of RNA using the iScript™ cDNA Synthesis Kit (Bio-Rad, Hercules, CA, USA) following the manufacturer’s instructions.

Real-time quantitative PCR (qPCR) reactions were performed in an iQ5 iCycler thermal cycler (Bio-Rad), setting a 96-well plate according to Chemello et al. [57]. For each sample, reactions were set mixing 1 μL of 1:10 diluted-cDNA, 5 μL of fluorescent intercalating agent (2× concentrated iQ ™ Sybr Green, Bio-Rad, Milano, Italy), and 0.3 μM of both forward and reverse primer. The thermal profile was 3 min at 95 °C, and then 45 cycles of 20 s at 95 °C, 20 s at the annealing temperature specific to each primer (reported in Table 2), and 20 s at 72 °C for extension. Annealing temperatures for each primer were optimized with temperature gradient assays. Primer specificities were assessed via the absence of primer–dimer formation and dissociation curves. Additionally, for each pair of primers, the efficiencies were evaluated with a mix of cDNA (Control group) (efficiency around 90% for all the primers, with an R^2^ that ranged from 0.994 to 0.998) at different concentrations (1:1, 1:10, 1:100, 1:1000). At the end of each cycle, the fluorescence was monitored, and one single peak was detected for each qPCR product in the melting curve analyses. For each reaction, two no template controls (NTCs) were added in each run to guarantee absence of contamination (no peaks were found for the NTC in each reaction). Amplification products were sequenced, and homology was verified. Internal reference genes, ribosomal protein L13 (*rpl13*) and actin-related protein 2/3 complex subunit 1A (*arpc1a*), were used to standardize the results. Relative quantification of genes involved in immune response (interleukin 1 beta, *il1b*; interleukin 10, *il10*; lipopolysaccharide-induced TNF factor, *litaf*) and oxidative stress (superoxide dismutase 1, *sod1*; superoxide dismutase 2, sod2; catalase, *cat*) was performed. Calculation of mRNA levels of target genes analysed was performed using the geometric mean of the two reference genes after demonstrating that they were stably expressed through applications implemented in the Bio-Rad CFX Manager 3.1. software. Gene transcript expression alterations among experimental groups are reported as relative mRNA abundance (arbitrary units) [58]. The qPCR data were processed using iQ5 optical system software version 2.0 (Bio-Rad), including GeneEx Macro iQ5 Conversion and GeneEx Macro iQ5 files. 

### 2.11. Statistical Analyses

All data were checked for normality (Shapiro–Wilk test) and homoscedasticity (Levene’s test). All the data were then analysed through one-way analysis of variance (ANOVA) followed by Tukey’s multiple comparison post hoc test, performed using the software package Prism 8 (GraphPad software version 8.0.2, San Diego, CA, USA). Significance was set at *p* < 0.05. 

## 3. Results

### 3.1. Biometry

As reported in Table 3, no significant differences were detected among all experimental groups in terms of SGR for both zebrafish larvae and juveniles.

### 3.2. Confocal Microscopy Allows for a Clear In Situ Visualization of MPs

No fluorescent MP microbeads were detected in all the tissues analysed for the Control, B50, and B500 groups at both the larval (Figure 1) and juvenile (Figure 2) stages. Polymer B microbeads were only found in the gut lumen (Figure 1b). Differently, polymer A microbeads were found in the intestine of both larvae and juveniles. Furthermore, while at 20 dpf (larval stage), polymer A microbeads were detected only at the intestinal level, at 60 dpf (juvenile stage), polymer A microbeads were also detected in the hepatic parenchyma and, to a lesser extent, at the muscular level, suggesting a time-dependent translocation of MPs among organs and tissues.

### 3.3. MPs Quantification

Table 4 reports the MPs quantification in whole larvae and in the intestine, liver, and muscle of juveniles. Regarding the larvae, no fluorescent microbeads were detected in the Control, B50, and B500 groups. Both the A50 and A500 groups showed the presence of fluorescent A microbeads, with A500 showing a significantly (*p* < 0.05) higher abundance compared to the A50 group. 

With regard to juveniles, no fluorescent microbeads were detected in the Control group in all the tissues analysed. Considering the intestine, group A showed a higher presence of MPs with respect to group B. Specifically, the A500 group showed the highest abundance of MPs when compared to the other experimental groups. The A50 group was characterized by a significantly (*p* < 0.05) higher abundance value compared to the B50 and B500 groups, which did not show significant differences between them. 

Considering the liver samples, the A500 group was characterized by a significantly (*p* < 0.05) higher abundance compared to the A50 group, while no fluorescent microbeads were detected in the B50 and B500 groups.

Finally, considering the muscle samples, the A500 group showed a significantly higher abundance compared to the A50 group, while no fluorescent microbeads were evident in the Control and B groups.

### 3.4. Histology

Considering both larvae (Figure 3a–e) and juveniles (Figure 3f–j), no structural alterations or signs of inflammation were evident in the hepatic parenchyma and in the intestine from all the experimental groups. 

Table 5 reports the histological indexes measured in intestine samples of larvae and juveniles. In particular, in both life stages, the B50 and B500 groups were characterized, in comparison to the Control, A50, and A500 groups, by (i) a significantly (*p* < 0.05) lower mucosal fold height; (ii) a higher relative abundance of Ab+ goblet cells.

### 3.5. Real-Time PCR Results

With respect to the larvae, no significant differences were detected among the experimental groups in the expression of genes involved in immune response (Figure 4a–c) and oxidative stress (Figure 4e,f), with the exception of the B500 group, which showed a significant sod1 downregulation compared to the A50 group (Figure 4d). 

With respect to juveniles, no significant differences were evident among the experimental groups in the expression of genes involved in immune response (Figure 5a–c). However, with regard to the expression of genes involved in oxidative stress response (Figure 5d–f), the A50 and A500 groups showed a significant (*p* < 0.05) *sod1*, *sod2*, and *cat* upregulation compared to the Control group, while in B50 and B500 groups, only *sod1* gene expression was significantly (*p* < 0.05) higher than the Control (Figure 5d). No significant differences were evident in *sod2* and cat gene expression (Figure 5e,f).

## 4. Discussion

Despite the large number of publications on MPs exposure in fish, information about the effects of different-sized polymers during the fish life cycle is still limited. While previous studies have demonstrated that different fish species are able to distinguish water-suspended MPs, enacting avoidance and rejection mechanisms [61,62,63,64], in the present study, both zebrafish larvae and juveniles ingested the tested fluorescent microbeads that were included in the feed during the preparation process. This result was clearly confirmed through the confocal microscopy and quantification analyses, which also suggested a different fate for the tested MPs in relation to their size and the fish life cycle stage analysed. Specifically, confocal microscopy analyses showed that polymer A microbeads were absorbed at the intestinal level in both zebrafish larvae and juveniles, while they were present at the hepatic and muscular level only in the juvenile stage. This result is in accordance with previous studies performed on adult zebrafish and goldfish, evidencing that MPs microbeads of 1–5 and 0.8 µm in size, respectively, were internalised by the fish intestinal villi and were detected at the hepatic level after at least 30–45 days of exposure [65,66]. This evidence suggests that, in the present study, the larval stage was not exposed to the dietary MPs (independent of the MPs’ dietary concentration) for a sufficient time to observe their translocation to other tissues and organs.

The MPs quantification through chemical digestion provided quantitative details about the MPs accumulation, evidencing, during the larval phase, a simple dose-dependent accumulation. In this phase, whole specimens were sampled because of the impossibility of isolating single organs and tissues due to their small size. However, more accurate results were obtained at the juvenile stage since the analyses were performed on single organs and tissues. At this developmental stage, polymer A microbeads were considerably present in the intestine and liver and, in a significantly lesser extent, in the muscle. These are extremely interesting results and are evidence of zebrafish possessing biological barriers capable of trapping MPs. Even though the precise mechanism via which MPs are absorbed at the intestinal level was not determined, different studies have proposed that the absorption paths could be via endocytosis or pinocytosis [67,68]. Then, once they are absorbed by the intestinal villi, MPs can reach different tissues through blood circulation [69]. In the present study, a similar scenario was possibly evidenced due to the higher number of MPs detected in the liver compared to the intestine. This result, along with the absence of inflammatory events, histopathological alterations, and changes in the expression of genes involved in immune response, suggests that the intestine acts as a simple “transit ”organ for dietary MPs. 

Conversely, the liver in zebrafish juveniles was able to accumulate a high number of polymer A microbeads, possibly constituting an “accumulation organ”. This result is also supported by the gene expression of oxidative stress markers that evidenced a strong upregulation (independent of the MPs’ dietary concentration) in the liver of zebrafish juveniles fed diets containing polymer A microbeads. Accordingly, previous studies highlighted that the liver is the organ most affected MPs exposure [70,71,72,73]. It also seems that MPs’ effects on the liver are strictly related to their size and the possible combination of the particles with toxic compounds, such as antibiotics [74] or metals [75].

Nevertheless, the ability of the liver to trap most of the polymer A microbeads ensured that only a limited number of them can be translocated to the muscle. The presence of this important hepatic biological barrier protects the muscle, and thus the edible portion, of the fish from dietary MPs contamination, offering a defence for potential final consumers. These results are in accordance with a previous study performed on European seabass, which highlighted that fluorescent MP beads of 1–5 µm in size were detected in the fillet in a significantly lower amount compared to the ingested quantity [34].

Previous studies highlighted that MPs characterized by a greater dimensional range of 100–400 µm and 20 µm, respectively, simply transit through the gut lumen without being absorbed [75,76]. Accordingly, in the present study, chemical digestion analyses highlighted an absence of polymer B microbeads during the larval stage, while a very low abundance was detected at the intestinal level in juveniles. However, no absorption of these MPs was detected through confocal microscopy at the intestinal level during the juvenile stage. The low abundance of polymer B microbeads in the intestine can be related to the 24 h starving period that preceded the sampling, a sufficient amount of time to have an empty gut. However, even if the polymer B microbeads were not directly absorbed at the intestinal level, both larvae and juveniles from groups B showed shortened mucosal folds and a higher abundance of Ab+ goblet cells compared to the other experimental groups. These results are likely related to an abrasive effect of polymer B microbeads at the intestinal level and to a consequent intensification of intestinal lubrication via an increase in goblet cells to facilitate the MPs’ expulsion, as demonstrated in previous studies [76,77]. However, no signs of intestinal inflammation or an increase in the expression of markers involved in the immune or oxidative stress response were observed in these fish, suggesting more adverse effects related to smaller MP microbeads compared to bigger ones. These molecular results can be considered accurate despite the absence of a non-reverse transcriptase control. In fact, primers were specifically designed to span an exon–exon junction (that only exists in cDNA) to exclude genomic DNA contamination. Additionally, RNA was treated through DNase and checked on agarose gel and through a NanoPhotometer. The molecular results obtained in the present study are in accordance with other studies that showed how nano-sized MPs can have more negative consequences than micron-sized ones for both fish and crustaceans [78,79]. These outcomes indicate that MPs’ effects on organisms are inversely related to their size.

Finally, growth was not affected by the dietary administration of both polymer types in zebrafish larvae and in juveniles. This result was partially in accordance with a previous study performed on zebrafish, in which growth was not affected by 20–27 µm dietary MPs during the larval stage (30 dpf) but was negatively affected after 90 and 360 days of exposure [80]. Another study, testing two different-sized MPs on spiny chromis *Acanthochromis polyacanthus,* evidenced that while MPs 2 mm in diameter did not affect fish growth, those with a size < 300 µm had a negative impact on fish growth [81]. As suggested by the authors of the previously cited study, fish growth could be influenced by MPs depending on different variables, such as their size, concentration, and shape, and thus further studies are necessary to better elucidate why no growth differences were detected in the present study.

## 5. Conclusions

The present study demonstrated that zebrafish have biological barriers against dietary MPs acting in relation to size, concentration, and exposure time, leading to different *scenarios* during the different fish life cycle stages. MP microbeads of 40–47 µm in size were not absorbed at the intestinal level and simply transited to the gut lumen, progressively causing a shortening of mucosal folds and an increase in mucous cells in both larvae and juveniles. On the other hand, MP microbeads of 1-5 µm in size were able to pass the intestinal barrier and, only in juveniles, translocate from the gut to other target organs and tissues, such as the liver and the muscle, in a dose-dependent way. However, the reduced amount of polymer A microbeads detected in the juveniles’ muscle samples indicated that the liver is a key organ in the retention of MPs. 

These results are important for the aquaculture sector but, at the same time, underline the need for further research to promote animal welfare by mitigating the negative side effects of MPs in fish.

## Figures and Tables

**Figure 1 animals-13-02256-f001:**
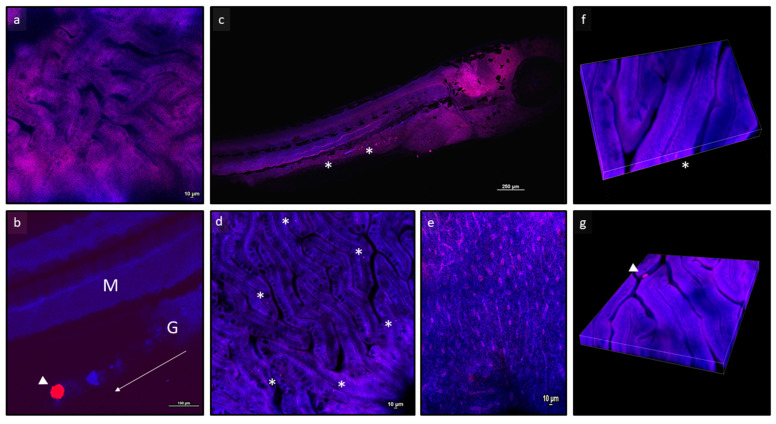
Representative images of zebrafish larvae analysed through confocal microscopy. (**a**) Focus on intestine from a zebrafish larva fed a control diet; (**b**) polymer B fluorescent microbeads in the gut lumen of a zebrafish larva fed a B500 diet; arrow indicates the direction of the gut tract, from cranial to caudal region; (**c**) whole larva fed an A50 diet; (**d**,**e**) focus on intestine and liver from larvae fed an A50 diet; (**f**,**g**) z-stack images of intestine from zebrafish larvae fed A50 and B50 diets, respectively. Asterisks indicate polymer A microbeads; arrowheads indicate polymer B microbeads. Abbreviations: M, muscle tissue; G, gut tract. * indicates microbeads.

**Figure 2 animals-13-02256-f002:**
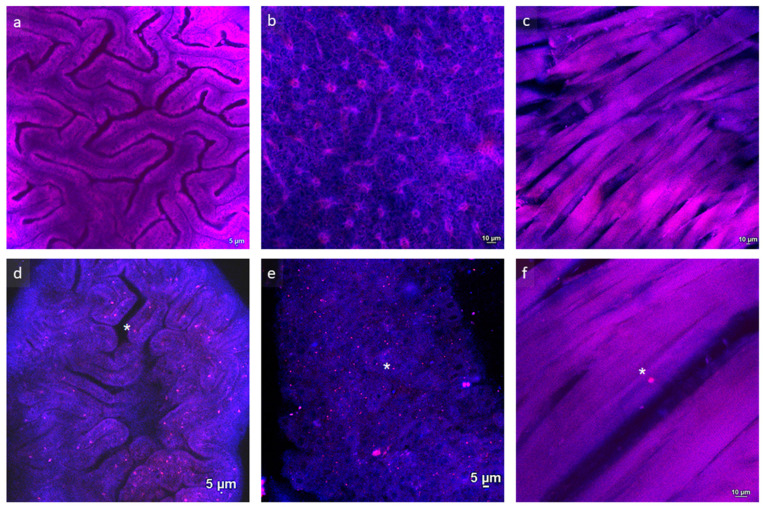
Representative images of (**a**,**d**) intestine, (**b**,**e**) liver, and (**c**,**f**) muscle samples of zebrafish juveniles fed Control (**a**–**c**) and A500 (**d**–**f**) diets. * indicates microbeads.

**Figure 3 animals-13-02256-f003:**
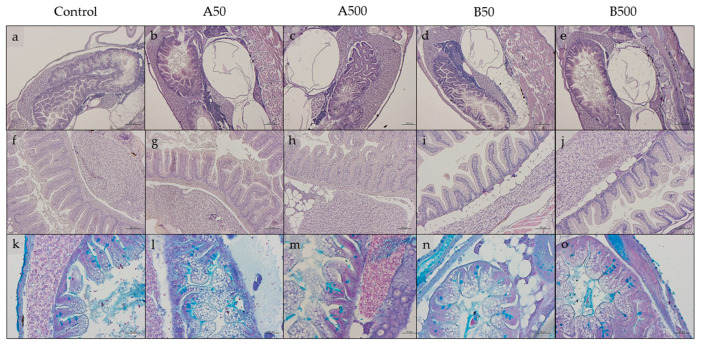
Examples of histomorphology of the gut tract and liver of zebrafish (**a**–**e**) larvae and (**f**,**g**) juveniles. (**k**–**o**) Examples of Ab+ goblet cells in intestinal mucosal folds. (**a**,**f**,**k**) Control; (**b**,**g**,**l**) A50; (**c**,**h**,**m**) A500; (**d**,**i**,**n**) B50; (**e**,**j**,**o**) B500. Scale bars: (**a**–**e**) 200 µm; (**f**–**g**) 100 µm; (**k**–**o**) 50 µm.

**Figure 4 animals-13-02256-f004:**
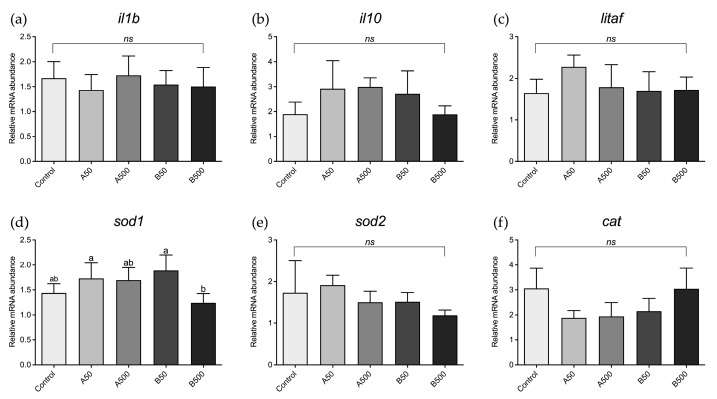
Relative mRNA abundance of genes involved in immune response (**a**) *il1b*, (**b**) *il10*, and (**c**) *litaf* and oxidative stress (**d**) *sod1*, (**e**) *sod2*, and (**f**) *cat* in analysed zebrafish larvae. Control: zebrafish fed a control diet; A50 and A500 groups: zebrafish fed diets containing 50 mg/kg and 500 mg/kg of polymer A (range size: 1–5 µm), respectively; B50 and B500 groups, zebrafish fed diets containing 50 mg/kg and 500 mg/kg of polymer B (range size: 40–47 µm), respectively. Data are reported as mean ± standard deviation (*n* = 5). ^a,b^ Different letters denote statistically significant differences among the experimental groups; *ns*, no significant differences.

**Figure 5 animals-13-02256-f005:**
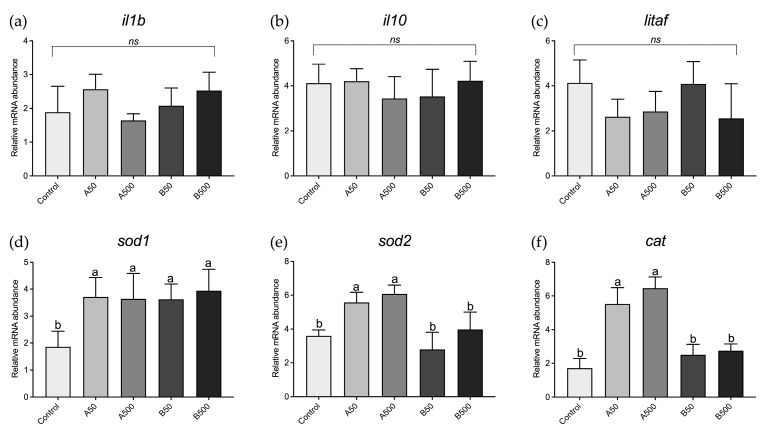
Relative mRNA abundance of genes involved in immune response (**a**) *il1b*, (**b**) *il10*, and (**c**) *litaf* and in oxidative stress (**d**) *sod1*, (**e**) *sod2*, and (**f**) *cat* analysed in intestine and liver, respectively, of zebrafish juveniles. Control: zebrafish fed a control diet; A50 and A500 groups: zebrafish fed diets containing 50 mg/kg and 500 mg/kg of polymer A (range size: 1–5 µm), respectively; B50 and B500 groups, zebrafish fed diets containing 50 mg/kg and 500 mg/kg of polymer B (range size: 40–47 µm), respectively. Data are reported as mean ± standard deviation (*n* = 5). ^a,b^ Different letters denote statistically significant differences among the experimental groups; *ns*, no significant differences.

**Table 1 animals-13-02256-t001:** Ingredients (g/kg), MPs concentrations (mg/kg feed), and proximate composition (% of DM) of experimental diets used in the present study.

	Control	Diet A50	Diet A500	Diet B50	Diet B500
**Ingredients (g/kg)**					
Fish meal ^1^	490	490	490	490	490
CPSP 90 ^2^	123	123	123	123	123
Wheat gluten meal ^3^	120	120	120	120	120
Pea protein concentrate ^4^	120	120	120	120	120
Wheat starch ^5^	55	55	55	55	55
Fish oil	60	60	60	60	60
Soya lecithin	8	8	8	8	8
Mineral and vitamin supplements ^6^	14	14	14	14	14
Binder (sodium alginate) ^7^	10	10	10	10	10
**MPs concentrations (mg/kg feed)**					
Polymer A (size: 1–5 µm)	-	50	500	-	-
Polymer B (size: 40–47 µm)	-	-	-	50	500
**Proximate composition (%)**					
Dry matter	94.2				
Crude protein	58.3				
Crude lipid	14.0				
Ash	12.5				

^1^ Fish meal (61% CP, 11% CF), kindly provided by Skretting Italia, Mozzecane (VR, Italy). ^2^ Soluble fish protein concentrate (82% CP) (Sopropêche, France). ^3^ Wheat gluten meal (CP, 81%), kindly provided by Skretting Italia. ^4^ Pea protein concentrate (CP 69%) (Lombarda trading srl, Cremona, Italy). ^5^ Wheat starch: pre-gelatinized wheat starch, kindly provided by Skretting Italia. ^6^ Mineral and vitamin supplement composition, as reported in [54]. ^7^ Sodium alginate (Merck KGaA, Darmstadt, Germany). For proximate composition, values are reported as mean ± standard deviation of triplicate analyses.

**Table 2 animals-13-02256-t002:** Sequences, identification numbers (ZFIN ID), and annealing temperatures (AT) of primers used in the present study.

Gene	Forward Primer (5′-3′)	Reverse Primer (5′-3′)	AT (°C)	ZFIN ID	Reference
*il1b*	GCTGGGGATGTGGACTTC	GTGGATTGGGGTTTGATGTG	54	040702-2	[59]
*il10*	ATTTGTGGAGGGCTTTCCTT	AGAGCTGTTGGCAGAATGGT	56	051111-1	[59]
*litaf*	TTGTGGTGGGGTTTGATG	TTGGGGCATTTTATTTTGTAAG	53	040704-23	[59]
*sod1*	GTCGTCTGGCTTGTGGAGTG	TGTCAGCGGGCTAGTGCTT	60	990415-258	[60]
*sod2*	CCGGACTATGTTAAGGCCATCT	ACACTCGGTTGCTCTCTTTTCTCT	60	030131-7742	[60]
*cat*	CCAAGGTCTGGTCCCATAA	GCACATGGGTCCATCTCTCT	60	000210-20	[60]
*rpl13*	TCTGGAGGACTGTAAGAGGTATGC	AGACGCACAATCTTGAGAGCAG	59	031007-1	[59]
*arpc1a*	CTGAACATCTCGCCCTTCTC	TAGCCGATCTGCAGACACAC	60	040116-1	[59]

**Table 3 animals-13-02256-t003:** Specific growth rate (SGR%) of zebrafish larvae and juveniles fed the experimental diets measured at 20 dpf and 60 dpf, respectively.

	Control	A50	A500	B50	B500	*p* Value
Larvae	15.2 ± 3.7	15.1 ± 3.6	16.1 ± 2.8	16.5 ± 2.6	15.9 ± 2.4	0.0603
Juveniles	9.4 ± 1.0	9.9 ± 1.6	9.7 ± 1.7	9.6 ± 1.5	9.5 ± 1.4	0.3861

Control: zebrafish fed a control diet; A50 and A500 groups: zebrafish fed diets containing 50 mg/kg and 500 mg/kg of polymer A (range size: 1–5 µm), respectively; B50 and B500 groups: zebrafish fed diets containing 50 mg/kg and 500 mg/kg of polymer B (range size: 40–47 µm), respectively. Data are reported as mean ± standard deviation (*n* = 60).

**Table 4 animals-13-02256-t004:** MPs quantification (number of microbeads/mg) in whole zebrafish larvae and in the intestine, liver, and muscle of zebrafish juveniles fed experimental diets.

		Control	A50	A500	B50	B500
Larvae	whole specimen	0	0.5 ± 0.2 ^a^	3.5 ± 0.8 ^b^	0	0
Juveniles	intestine	0	1.15 ± 0.45 ^a^	61.93 ± 14.30 ^b^	0.14 ± 0.01 ^a^	0.64 ± 0.15 ^a^
	liver	0	5.4 ± 1.6 ^a^	231.1 ± 47.1 ^b^	0	0
	muscle	0	0.3 ± 0.1 ^a^	4.7 ± 1.2 ^b^	0	0

Control: zebrafish fed a Control diet; A50 and A500 groups: zebrafish fed diets containing 50 mg/kg and 500 mg/kg of polymer A (range size: 1–5 µm), respectively; B50 and B500 groups, zebrafish fed diets containing 50 mg/kg and 500 mg/kg of polymer B (range size: 40–47 µm), respectively. Data are reported as mean ± standard deviation (*n* = 9). ^a,b^ Within each line, different letters denote statistically significant differences among the experimental group.

**Table 5 animals-13-02256-t005:** Histological indexes measured in the intestine of larvae and juveniles fed experimental diets.

		Control	A50	A500	B50	B500
Larvae	Mucosal fold height	102.9 ± 15.0 ^a^	86.7 ± 8.4 ^ab^	88.0 ± 5.8 ^ab^	73.2 ± 4.6 ^bc^	65.7 ± 6.0 ^c^
	Ab+ goblet cells’ relative abundance	+	+	+	++	++
Juveniles	Mucosal fold height	94.9 ± 5.7 ^a^	96.4 ± 8.8 ^a^	88.2 ± 9.4 ^a^	69.7 ± 7.9 ^b^	70.1 ± 5.4 ^b^
	Ab+ goblet cells’ relative abundance	++	++	++	+++	+++

Control: zebrafish fed a control diet; A50 and A500 groups: zebrafish fed diets containing 50 mg/kg and 500 mg/kg of polymer A (range size: 1–5 µm), respectively; B50 and B500 groups, zebrafish fed diets containing 50 mg/kg and 500 mg/kg of polymer B (range size: 40–47 µm), respectively. Data are reported as mean ± standard deviation (*n* = 15). ^a,b,c^ Different letters denote statistically significant differences among the experimental groups. Ab+ goblet cells: + = 0 to 3 per villus; ++ = 4 to 6 per villus; + + + = more than 6 per villus.

## Data Availability

The data presented in this study are available on request from the corresponding author.

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
