# Peer review of "Dietary Microplastic Administration during Zebrafish (Danio rerio) Development: A Comprehensive and Comparative Study between Larval and Juvenile Stages"

_animals, 2023, doi:10.3390/ani13142256_

Round 1

Reviewer 1 Report

Dear Nico Cattaneo and co-authors,

Thank you for your highly significant and interesting study. It was a pleasure to read and review your manuscript titled "Dietary microplastic administration during zebrafish (Danio rerio) development: a comprehensive and comparative study between larval and juvenile stages". Your experiment design is thoughtful and well done. With your high sample size and replicates per group, you build up a highly significant study and thus contribute highly to the MP research field. Your results are presented well-structured, joined by reliable results and backed up with comparable studies. 

I would only like to request two things:

- please, describe in line 191 how you stored your samples. This is one of the most critical points and thus needs to be presented. Information about sample container/bag/aluminium foil, -20°C or -80°C needs to be added. 

- please, write out the definition of PFA in L207, when you use it the first time since it needs to be clarified, even if it can be obvious.

Thank you for your effort with this significant study.  

Author Response

Reviewer 1

Dear Nico Cattaneo and co-authors,

Thank you for your highly significant and interesting study. It was a pleasure to read and review your manuscript titled "Dietary microplastic administration during zebrafish (Danio rerio) development: a comprehensive and comparative study between larval and juvenile stages". Your experiment design is thoughtful and well done. With your high sample size and replicates per group, you build up a highly significant study and thus contribute highly to the MP research field. Your results are presented well-structured, joined by reliable results and backed up with comparable studies. 

I would only like to request two things:

- please, describe in line 191 how you stored your samples. This is one of the most critical points and thus needs to be presented. Information about sample container/bag/aluminium foil, -20°C or -80°C needs to be added. 

- please, write out the definition of PFA in L207, when you use it the first time since it needs to be clarified, even if it can be obvious.

Thank you for your effort with this significant study. 

The authors would like to thank the reviewer for the positive comments about the present MS. The two requests have been now satisfied. Particularly, information about samples storage has been added in the dedicated section of the revised MS. In this sentence, PFA and PBS  have  been clarified.

Reviewer 2 Report

The study investigated a comparative approach based on fish developmental stage (larval vs juvenile) and exposure time, dietary MPs size and concentration for zebrafish (Danio rerio). The experimental setup is relatively simple, and the effect of different concentrations of larger and smaller microplastics on different stages of zebrafish is relatively examined, but each part of the study is not very deep, only to show the phenomenon. What is more, some tables in the paper suggest that line charts or bar charts can be used to replace the results, which will be more intuitive. In general, there are no serious problems with this article, and it is recommended to accept after minor modification.

 Introduction

Ln 43-44: Just “been detected in several environments” is fine, and remember to change the quote from [1-3] to [1-4].

Ln 50: “including fish” is not necessary, it can be deleted.

Ln 67: “anatomical” can be replaced by “histology”, and the same below.

Ln 83: Does “fillet” refer to the muscle part of the fish?

Ln 107: “the larval to the juvenile stage”.Please explain in the Introduction how these two stages are distinguished and their difference.

Materials and Methods

Ln 131: How do you make feed without microplastics?

Ln 166: What do you select based on? Please explain in the text.

Ln 172: How much water is replaced each time?

Ln 186-187: Doesn't the feed start at 5 dpf? Why do you feed rotifers at 5-10 dpf? Then how many rotifers were fed and how many times? 

Ln 192: Why is the number of fish tested different at each stage?

Ln 199: “17 and 57 for larvae and juveniles” Should not be “20 and 60 for larvae and juveniles”?

Ln 242-243: “performed on three transversal sections per fish collected at 50 μm intervals.” I don't quite understand the meaning of this sentence.

Results Ln 304: This headline suggestion could be replaced with a more appropriate statement.

Discussion

Ln464-468: As you say that “the MPs effects on the organisms are inversely related to their size”, then why “nano-sized MPs can have more negative consequences than micron-sized ones”? Aren't nano-sized MPs (nm) bigger than micron-sized ones(μm)?

Conclusions

Ln 488-490: Why to say “the reduced amount of polymer A microbeads detected in the juveniles’ muscle samples indicated that the liver is a key organ in retaining the MPs”? What is the connection between muscle and liver?

Author Response

Reviewer 2

The study investigated a comparative approach based on fish developmental stage (larval vs juvenile) and exposure time, dietary MPs size and concentration for zebrafish (Danio rerio). The experimental setup is relatively simple, and the effect of different concentrations of larger and smaller microplastics on different stages of zebrafish is relatively examined, but each part of the study is not very deep, only to show the phenomenon. What is more, some tables in the paper suggest that line charts or bar charts can be used to replace the results, which will be more intuitive. In general, there are no serious problems with this article, and it is recommended to accept after minor modification.

Thank you for the suggestion. However, the authors suggest maintaining tables, especially for MPs quantification. Creating a panel with several graphs for each tissue, for both larvae and juveniles, could be more dispersive and not easily comparable. We thus hope that the tables can be maintained, also in light to the absence of comments on this topic from other reviewers. In case many thanks.

 Introduction

Ln 43-44: Just “been detected in several environments” is fine, and remember to change the quote from [1-3] to [1-4].

Thanks, Done

Ln 50: “including fish” is not necessary, it can be deleted.

Deleted

Ln 67: “anatomical” can be replaced by “histology”, and the same below.

The term “anatomical” is more correct for referring to changes in body structure of organisms. Histology is just a laboratory technique to assess the anatomical structures. For this reason we would prefer to leave the sentence as it is.

Ln 83: Does “fillet” refer to the muscle part of the fish?

Yes it does. To better clarify, “fillet” has been replaced by “muscular tissue”. Thanks for the suggestion.

Ln 107: “the larval to the juvenile stage”.Please explain in the Introduction how these two stages are distinguished and their difference.

A sentence (line 100 of the revised MS) has been added to clarify this aspect.

Materials and Methods

Ln 131: How do you make feed without microplastics?

Control diet was the first diet produced, without any contamination derived from the fluorescent MPs used in the present study (MPs were in fact bought form a company, manually added during the preparation of the diets and thus if they were not added to the diet mixture these MPS were not present). Additionally, to be completely sure of the absence of contamination, the presence of fluorescent MPs was checked through confocal microscopy in all diets as reported in line 142 of the revised MS. As stated in the MS, no fluorescent MPs were detected in control diet. In each case we added a sentence into the MS to clarify this point.

Ln 166: What do you select based on? Please explain in the text.

We selected the embryos that were correctly developing and not damaged by observing them under the microscope, removing dead and damaged ones. A sentence has been added

Ln 172: How much water is replaced each time?

The entire volume of water of the tank (20L) was replaced by the dripping system 10 times a day. (20Lx 10 times= 200Lday)

Ln 186-187: Doesn't the feed start at 5 dpf? Why do you feed rotifers at 5-10 dpf? Then how many rotifers were fed and how many times? If you are co-feeding rotifers, and larvae are eating them, the concentration of rotifers will be reduced along time. How did you maintain the same level?

The feed was provided from 5 dpf since at this point the exougenous feeding of larvae starts. However, fish in the early larval stages cannot rely only on dry feed; thus, a supplementation of live prey is recommended for a proper fish development (please see reference reported). 5 rotifers/mL were provided two times a day (one in the morning and one the afternoon) for each tank.

Actually, there is no need to maintain the whole day a 5 rotifer/mL concentration in the tank since larval fish are fed two or more times a day. This is a common practice in all larval fish rearing. With this interval fish have time to finish the rotifers, that will not pollute the water, but receive them again on time without starving. A clarifying sentence has been added.

Ln 192: Why is the number of fish tested different at each stage?

The higher number of larvae required for some analyses (i.e. molecular analyses) compared to juveniles organs and tissues  is mainly due to the larval small size that were often pooled to achieve the correct amount of sample necessary for a certain type of analysis.

Ln 199: “17 and 57 for larvae and juveniles” Should not be “20 and 60 for larvae and juveniles”?

For the SGR calculation, the initial body weight was measured at 3 days post fertilization, when the hatching occurred. For that reason, from the total number of days of the trial (20 and 60 for larvae and juveniles, respectively), the 3 days of embryo development must be removed (thus obtaining 17 and 57, respectively).

Ln 242-243: “performed on three transversal sections per fish collected at 50 μm intervals.” I don't quite understand the meaning of this sentence.

The sentence has been modified as follows: “The evaluation of histological indexes in the intestine was performed on three transversal sections per fish (15 fish per dietary group) collected at a 50 μm distance from each other”.

Results Ln 304: This headline suggestion could be replaced with a more appropriate statement.

The headline has been replaced with “growth performance”

Discussion

Ln464-468: As you say that “the MPs effects on the organisms are inversely related to their size”, then why “nano-sized MPs can have more negative consequences than micron-sized ones”? Aren't nano-sized MPs (nm) bigger than micron-sized ones(μm)?

No they aren’t. Nano-sized MPs (nm) are three order of magnitude smaller than micron-sized MPs (1 μm = 1 x 10-6 m ; 1 nm = 1 x 10-9 m)

Conclusions

Ln 488-490: Why to say “the reduced amount of polymer A microbeads detected in the juveniles’ muscle samples indicated that the liver is a key organ in retaining the MPs”? What is the connection between muscle and liver?

There is not a direct connection between muscle and liver, but all the tissues are “in communication” through the circulatory system. As reported in the discussion, it has been demonstrated that liver is a key organ in trapping MPs, thus removing these contaminants from the blood stream, and preventing their deposition in other tissues, like muscle.